# Stilbene Compounds Inhibit Tumor Growth by the Induction of Cellular Senescence and the Inhibition of Telomerase Activity

**DOI:** 10.3390/ijms20112716

**Published:** 2019-06-02

**Authors:** Yu-Hsuan Lee, Yu-Ying Chen, Ya-Ling Yeh, Ying-Jan Wang, Rong-Jane Chen

**Affiliations:** 1Department of Food Safety/Hygiene and Risk Management, College of Medicine, National Cheng Kung University, Tainan 70428, Taiwan; bmm175@gmail.com; 2Department of Environmental and Occupational Health, College of Medicine, National Cheng Kung University, Tainan 70428, Taiwan; 101312123@gms.tcu.edu.tw (Y.-Y.C.); linn7627@hotmail.com (Y.-L.Y.); 3Department of Medical Research, China Medical University Hospital, China Medical University, Taichung 40402, Taiwan

**Keywords:** senescence, telomerase, stilbene compounds, resveratrol, pterostilbene

## Abstract

Cellular senescence is a state of cell cycle arrest characterized by a distinct morphology, gene expression pattern, and secretory phenotype. It can be triggered by multiple mechanisms, including those involved in telomere shortening, the accumulation of DNA damage, epigenetic pathways, and the senescence-associated secretory phenotype (SASP), and so on. In current cancer therapy, cellular senescence has emerged as a potent tumor suppression mechanism that restrains proliferation in cells at risk for malignant transformation. Therefore, compounds that stimulate the growth inhibition effects of senescence while limiting its detrimental effects are believed to have great clinical potential. In this review article, we first review the current knowledge of the pro- and antitumorigeneic functions of senescence and summarize the key roles of telomerase in the regulation of senescence in tumors. Second, we review the current literature regarding the anticancer effects of stilbene compounds that are mediated by the targeting of telomerase and cell senescence. Finally, we provide future perspectives on the clinical utilization of stilbene compounds, especially resveratrol and pterostilbene, as novel cancer therapeutic remedies. We conclude and propose that stilbene compounds may induce senescence and may potentially be used as the therapeutic or adjuvant agents for cancers with high telomerase activity.

## 1. Introduction

Cellular senescence is a stable cell cycle arrest that occurs in diploid cells that is caused by a progressive shortening of telomeres during each cell division. This phenomenon is currently defined as replicative senescence [1]. However, diploid cells can also experience an accelerated senescence response that is independent of the telomere shortening and that occurs immediately after genotoxic stress, metabolic shock, oncogenic activation, or the loss of tumor suppressor genes in primary and tumor cells [2,3]. For instance, several anticancer chemotherapies and radiotherapies are known to induce senescence in both normal and cancer cells [4]. Since the discovery of telomerase, which is an enzyme that is able to elongate telomeres and is mostly active in tumors, it has become a prominent target of therapy. Although different methods of inhibiting telomerase have been attempted in the preclinical studies, no directs telomerase inhibitor has been approved for clinical use so far [5,6,7]. There are several different components of telomerase, including the catalytic subunit telomerase reverse transcriptase (hTERT), the RNA component hTR, and associated proteins. In addition, telomeres also contain special structures and associated proteins and assume a variety of conformations, all of which contribute to the activity of telomerase [8]. Previous studies have demonstrated that senescence occurs in different tumor tissues, where it attenuates tumor development and progression. Thus, according to its antiproliferative effects, senescence is considered to be a potent antitumor mechanism. The tumor-suppressive function of senescence provides an opportunity for the development of treatments that enhance senescence in clinical cancer therapy [1]. Moreover, recent emerging evidence has indicated that immune responses against senescent cells are crucial to restricting disease progression in cancerous pathologies. If the senescent cancer cells in treated tumor tissue cannot be completely eliminated by a proper senescence immune surveillance response, the treatment is failing [9,10]. Despite their involvement in various pathological conditions, senescent cells also play pivotal roles in normal physiological processes such as embryogenesis, tissue remodeling, and tissue repair [11].

Cancer cells have been demonstrated to be forced to undergo senescence by natural compounds, with similar effects to those obtained by genetic and epigenetic manipulations, anticancer drugs, and irradiation. Thus, the use of cellular senescence to drive towards antitumor adjuvant therapies by natural compounds is currently attracting interest [12]. Examples of these natural bioactive components include stilbenes, which are a subgroup of polyphenols that occur in many plant species [13]. In light of their safety profiles in humans, it is suggested that stilbene compounds can potentially be used for chemotherapy or antitumor adjuvant therapies. In this review, we provide an overview of the causes of cellular senescence with a particular focus on the stilbene compounds that have been shown to induce senescence and inhibit telomerase activity in cancer cells.

## 2. Senescence in Cancer Cells and Its Mechanisms

### 2.1. Characteristics of Senescence

Cellular senescence is a state of cell cycle arrest that happens in proliferating cells that are subjected to different stresses which serves as a cellular protection mechanism that helps cells to avoid unnecessary damage. Senescence occurs in several tissues during different physiological and pathological processes, such as tissue remodeling, injury repair, cancer, and aging [14]. Current studies have indicated that senescence is a stress response triggered by several mechanisms, such as telomere shortening, the accumulation of DNA damage, and oxidative stress [15]. These mechanisms are involved in protection against cancer and are also involved in organismal senescence [16]. Senescent cells exhibit permanent growth arrest, the increased expression of cell cycle inhibitors, changes in cellular structures, and protein expression. In senescent cells, cell cycle arrest correlates with increased levels of cell cycle inhibitors, including p16^INK4a^, p21^CIP1/WAF1^, and p27^Kip1^. Moreover, elevated expression of p19ARF, p53, and PAI-1is observed in senescent cells serve as senescence biomarkers [14].

### 2.2. Effector Signaling Pathways Involved in Senescence

Senescence occurs in the contexts of a number of different stresses. Among them, telomere shortening represents one of the most important outcomes. Telomeres are repetitive nucleotide-sequence motifs that protect the ends of chromosomes from degeneration or fusion with neighboring chromosomes. Each cell division leads to the loss of 50–200 bp of unreplicated DNA at the 3′ end of the telomere. The enzyme telomerase (also known as terminal transferase) is responsible for adding bases to the end of telomeres to compensate for telomere erosion [5]. Whenever telomerase activity is not sufficient to keep up the rate of cell proliferation, the results is telomere shortening and cell senescence. Another cause of senescence is DNA damage, during which ataxia-telangiectasia mutated (ATM) or the ATM- and Rad3-related (ATR) kinases block cell-cycle progression via stabilization of the p53 protein and the transcriptional activation of the cyclin-dependent kinase (Cdk) inhibitor p21^CIP1/WAF1^ [17]. Senescent cells with DNA damage contain positive γ-H2AX (a phosphorylated form of the histone variant H2AX) and the DNA damage response (DDR) proteins 53BP1, NBS1, and MDC1. These molecular events can induce a transient proliferation arrest that can evolve into senescence if cells are not able to repair the damage. Oxidative stress also participates in telomere erosion and DNA damage. Moreover, endoplasmic reticulum stress or interferon (IFN)-related responses induce cellular senescence [18]. Senescence can also be triggered by the activation of oncogenes (oncogene-induced senescence (OIS)) and the loss of tumor suppressor genes (TSGs), as discussed in detail in the following section.

### 2.3. Senescence in Cancer Cells

Oncogene activation results in proliferation and senescence induction limits tumor growth. A well-known example of the evidence of OIS is the overexpression of RAS. Mutations in the RAS oncogene are common in many human cancers. In the absence apoptosis, stressors could also drive cells into senescence through RAS, and this mechanism works as a barrier to block tumor growth in vivo [19]. OIS is accompanied by the concomitant upregulation of p19ARF, Pml, p53, retinoblastoma (Rb) protein, and p16^INK4a^, and the inactivation of these genes results in the evasion of HRASG12V-induced cellular senescence [20].

Several drugs in clinical use for the management of human cancers can mediate therapy-induced senescence (TIS), including docetaxel, bleomycin, cyclophosphamide, doxorubicin, vincristine, etoposide, and cisplatin [4]. The mechanisms that force tumor cells into senescence are generally linked to the enhancement of DNA damage. Primary murine lymphomas have been shown to respond to chemotherapeutic treatment with cyclophosphamide by engaging a senescence program controlled by p53 and p16^INK4a^ [21]. Several targeted therapies that inhibit CDKs, NOTCH, CK2, MDM2, JAK2, and SKIP2 can also promote growth arrest and senescence in tumors [22].

## 3. Senescence Is a Novel Target for Anticancer Therapy

Although the regulators and pathways involved in anticancer therapy mediated by cellular senescence are complex and the detailed mechanisms still need to be elucidated, several vital processes have been well described in recent years, such as those involved in the regulation of telomerase, DNA replication, DNA damage, permanent cell cycle arrest, and the mitotic inhibition [23]. The action of some clinical cancer drugs in regard to cellular senescence and the latest TIS strategies are discussed in the following paragraph.

### 3.1. Telomerase Inhibition by Chemotherapeutic Drugs

Paclitaxel is a chemotherapeutic drug, isolated from the bark of the Pacific yew *Taxus brevifolia* that has been used to treat a number of types of cancer. Paclitaxel is the first microtubule stabilizing agent that could suppresses spindle microtubule dynamics, resulting in the inhibition of mitosis and induction of apoptosis in cancer cells. In addition, paclitaxel and its water-soluble conjugates can inhibit tumorigenesis by inducing extensive telomere erosion [24]. These drugs can also stimulate chromosomal fusion and instability in cells with dysfunctional telomerase [25]. Bleomycin (BLM) is another chemotherapeutic drug isolated from *Streptomyces verticillus* commonly used to treat in different kind of cancers, such as lung cancer, cervical cancer, and cancers of the head and neck. Studies have indicated that BLM induces not only single- and double-strand breaks in DNA [26,27] but also the persistent loss of chromosome ends and telomere dysfunction to inhibit tumor growth [28]. Because telomerase mutations that cause cell senescence are genetic risk factors for the occurrence of many cancers, the targeting of telomerase is a promising strategy for anticancer therapy (we will discuss the effects of telomerase on anticancer therapy in more detail in Section 3).

### 3.2. DNA Damage Triggered Senescence

Chemotherapeutic drugs can also be designed that affect DNA replication and promote DNA damage, leading to senescence. In general, the proper regulation of DNA replication ensures the faithful transmission of genetic material to daughter cells and the maintenance of genomic stability in cell proliferation. However, this highly regulated process can be disrupted when DNA replication proceeds in cancer cells with elevated rates of genomic instability and increased proliferative capacities [29,30]. Recently, a small molecule inhibitor of the checkpoint kinase CHK1, which mediates cell-cycle arrest to facilitate DNA repair [31], was reported to be in clinical development in combination with the antimetabolite gemcitabine. Gemcitabine is an analogue of cytosine arabinoside (Ara-C) [32] and a standard treatment for patients with pancreatic ductal adenocarcinoma. Song et al. showed that gemcitabine significantly increased the levels of senescence-associated molecules, including p53, p21^CIP1/WAF1^, p19ARF, PML, and DCR2, in Miapaca-2 and Panc-1 cells, with the exception of p53 in Panc-1 cells [33]. Therefore, combining cell-cycle checkpoint kinase inhibitors with the DNA-damaging chemotherapeutic agents has clinical appeal because the inhibition of the DDR with checkpoint kinase inhibitors and the induction of cellular senescence will enhance the chemosensitization of p53-mutant pancreatic cancer cells [34]. DNA-damaging agents are well known to induce senescence in tumor cells; however, most standard genotoxic chemotherapeutic regimens have proven unsuccessful in patients. One reason for this is that tumor cells develop resistance to DNA-damaging chemotherapeutic agents by acquiring the ability to repair their DNA. Combination therapies that induce DNA damage and disrupt the DNA damage repair mechanisms were investigated in a recent study. For instance, the transcription factor TBX2 has been suggested as a novel anticancer drug target. The overexpression of TBX2 produces strong antisenescence and proliferative effects. The knock down of TBX2 enhanced the effects of cisplatin by disrupting the ATM-CHK2-p53 signaling pathway in a cisplatin-resistant breast cancer cell line [35]. These studies indicate that the induction of senescence in response to chemotherapy-induced DDR could produce better outcomes for patients with cancer. 

### 3.3. Permanent Cell Cycle Arrest by Cell Cycle Regulators

Chemotherapeutic drugs can also trigger cellular senescence by permanent cell cycle arrest. The cell cycle is controlled by a family of protein kinases known as CDKs and by cyclins [36,37]. Sustained activation of CDK inhibitors, such as p16^INK4a^, p15, p27^Kip1^, and p21^CIP1/WAF1^, induces senescence in experimental model systems [20,38]. This observation had brought up the idea that drugs able to enhance the levels of CDK inhibitors or inhibit CDKs can be used to induce senescence as a tumor suppression mechanism. Dexamethasone is a type of corticosteroid medication. It is used in the treatment of many conditions. In pemetrexed-based chemotherapy, dexamethasone is co-administered to alleviate severe drug-induced and painful skin rashes [39,40]. In addition, Patki et al, utilized dexamethasone to explore the vital role of p27^Kip1^ in cellular senescence, and the results showed that long-term dexamethasone treatment induced irreversible cell cycle blockade and a senescence phenotype through chronic activation of the p27^Kip1^ in glucocorticoid receptor-α (GR)-overexpressing lung tumor cells [41]. The modulation of the p21^CIP1/WAF1^ activity has been shown to have a similar effect in inducing the senescence response. Prostate apoptosis response-4 (Par-4), which is a proapoptotic tumor suppressor protein, is unique in its ability to selectively induce apoptosis, cell cycle arrest, and increase the expression of p53 and p21^CIP1/WAF1^ in cancer cells [42,43]. Other drugs that are currently being investigated for the ability to induce senescence by modulating the cell cycle machinery are the CDK inhibitors. For example, a CDK4/6-specific inhibitor, palbociclib (also known as PD0332991), was approved by the FDA for the treatment of advanced estrogen receptor-positive breast cancer [44]. Other CDK4/6-specific inhibitors, such as ribociclib (LEE011) and amebaciclib (LY2835219), are both in phase I–II clinical trials and they have been tested alone or in combination with chemotherapy in different cancer cells [45,46].

### 3.4. Mitotic Inhibition-Induced Senescence

Several mitosis-specific agents, such as inhibitors of mitotic kinase and kinesin spindle proteins, have been designed and are undergoing evaluation and research. Currently, mitosis-based treatment is a validated approach for the treatment of non-small cell lung cancer (NSCLC). Mitosis-targeted drugs can be categorized as one of several types, including microtubulin binders, microtubulin enzyme inhibitors, mitosis checkpoint kinase (CHK) inhibitors, and mitosis enzyme inhibitors. The natural-source compounds, paclitaxel and taxotere (docetaxel), are well-known members of taxanes that act as microtubulin binders. The primary cytotoxic mechanisms of paclitaxel are the stabilization of GDP-bound tubulin in microtubules and the disruption of microtubule dynamics, resulting in the blockage of cell division during mitosis and senescence [47,48,49,50]. LY2603618 is a selective and potent ATP-competitive CHK1 inhibitor. This type of mitotic-specific agent impairs DNA synthesis, increases DNA damage by producing mitotic defects, and induces apoptosis, autophagy, and senescence in cancer cells [51,52,53]. A Phase I clinical study reported that LY2603618 showed acceptable safety in combination with pemetrexed and cisplatin [54]. Moreover, the use of mitotic enzyme inhibitors such as polo-like kinase (PLK) inhibitors is also a promising therapeutic strategy via the arrest of cells in mitosis. PLK1 plays a role in centrosome disjunction and separation and is highly expressed in human cancers. Thus, PLK1 inhibition by various agents can cause postmitotic DNA damage and senescence in several cancer cell lines [55,56,57].

### 3.5. The Latest Therapy-Induced Senescence (TIS) Strategies for Cancer Cells

Traditional cancer therapy attempts to kill cancer cells by inducing extensive DNA damage using high doses of drugs or irradiation, but this kind of treatment causes severe side effects during cancer therapy. To solve this problem, the use of TIS, which is a senescent response caused by conventional cancer therapy, is suggested as a strategy for overcoming these side effects. In general, there are three typical strategies for TIS that can be considered: (i) the activation of DDR-related senescence by radiation or genotoxic chemotherapeutics, (ii) the reactivation of senescence-mediated tumor suppressor genes, and (iii) the induction of cytokine-induced senescence (CIS). Hinds and Pietruska have collated the relevant content in a previous literature review [58]. For the induction of senescence in tumor cells by genotoxic stress or the reactivation of growth-suppressive pathways, the reactivation or re-expression of p53, the inactivation of Myc or Bcr-Abl, and the inhibition of Pten are attractive therapeutic approaches and have been shown to potentiate growth inhibition or tumor regression [59,60,61]. In contrast to cell-autonomous mechanisms of TIS that are induced by genotoxic stress or the reactivation of growth-suppressive pathways, CIS limits tumor cell proliferation through immunity-induced senescence, which triggers an increase in T_H_1-cytokines and a stable cell cycle arrest in cancer cells. A recent study has shown that the T_H_1 cytokines interferon gamma (IFN-γ) and tumor necrosis factor (TNF) induce senescent growth arrest in vitro that is accompanied by restoration of functional p16^INK4A^/pRb [62]. In addition, Rentschlera et al. demonstrated that both IFN-γ and TNF can induce senescence in MCF-7 breast cancer cells and A204 rhabdomyosarcoma cells, resulting in reduced tumor cell proliferation [63].

Although TIS is a well-established response to conventional cancer therapy, there is now extensive evidence indicating that TIS might be transient and eventually cause disease recurrence. Biologists are concerned that TIS could represent an undesirable outcome of therapy by providing a mechanism for tumor dormancy. Recently, the senescence-associated secretory phenotype (SASP) has been proposed as a major driver of escape from senescence and the re-emergence of proliferating tumor cells. Therefore, targeting and inhibiting SASP might serve to mitigate the deleterious outcomes of TIS. [64]. Although several small molecules and compounds that could be used in senescence therapy are currently undergoing clinical trials, senescent cells cannot be cleared by immune cells after treatment. Accordingly, the development of senolytic drugs that can induce cell death in senescent cells are gaining increasing attention [65,66,67]. It is expected that the combination of senolytic drugs and TIS may ultimately be of great clinical benefit for cancer therapy [68].

## 4. Targeting of Telomerase as an Attractive Anticancer Strategy

The ectopic subunit of the telomerase complex is able to maintain the telomere length and immortalize normal cells. Nearly all of the human tumors have been shown to be telomerase-positive, and malignant tumors with the capacity for unlimited cell proliferation are all characterized by the overexpression of telomerase. Mutation of the proximal promoter of the human TERT gene () is now considered to be the most common noncoding mutation in cancer. It is believed that such mutations activate telomerase activity by converting conserved regions within the TERT promoter into an ETS transcription factor, binding sites to permit the continuous cell divisions that are required for advanced cancers [69]. Telomerase plays an important role in cancer cell proliferation and is highly overexpressed in 85% of tumors but is almost undetectable in most normal tissues [70]. Therefore, telomerase is suitable as an anticancer target. Studies have shown that telomerase inhibitors have higher specificity and fewer side effects than traditional cancer chemotherapy [71]. Telomerase-based therapeutic approaches treat cancer by directly inhibiting telomerase activity or causing changes in the gene expression level. The inhibition of telomerase resulted in a decrease in telomere length, which then prompted tumor cells to undergo senescence and apoptosis [72,73]. 

### 4.1. Inhibition of Telomerase Components in Cancer Therapy

Telomerase is mainly consisted of the catalytic subunit hTERT, which is essential for telomerase activity, and the RNA moiety hTR. The most intuitive strategy is designed to directly target the core of telomerase components; both hTERT and hTR have been used as targets for telomerase inhibition. Azidothymidine (AZT), which is the first drug used to treat AIDS and HIV infection and is also known as zidovudine (ZDV), was the first reported telomerase inhibitor. Several studies have indicated that, because HIV reverse transcriptase and telomerase are similar, AZT preferentially binds to the telomeric region of DNA in CHO cells [74,75]. Next, some studies have reported that AZT can enhance senescence and apoptosis and decrease cell proliferation via the inhibition of telomerase and the reduction in telomere length in immortal cells [76,77]. Accordingly, AZT has been used in anticancer therapy, and has been subjected to phase I and II clinical trials. 

In fact, several clinical anticancer medications have also been shown to have the antitelomerase capacity. Here, we list only a few examples, which are described below. Cisplatin, a well-known anticancer drug used to treat many types of cancer, has been shown to inhibit telomerase in HeLa and hepatoma cells [78,79]. In this study, Zhang et al. indicated that cisplatin induced the accumulation of cancer cells in the G2/M of the cell cycle phase and the inhibition of cell growth by decreasing hTERT mRNA. Furthermore, a number of new antitelomerase drugs are currently in development. For example, BIBR1532 (2-[E)-3-naphtalen-2-yl-but-2-enylylamino]-benzoic acid), which is a synthetic small molecule telomerase inhibitor, is one of the most successful hTERT inhibitors [80,81]. BIBR1532 acts by binding to hTERT, which contains the active site of telomerase, and directly inhibiting the enzyme [82]. Many studies reported that BIBR1532 has strong effects on telomere shortening, cell growth arrest, senescence, and apoptosis in several human cancer cell lines, including prostate cancer, breast cancer, fibrosarcoma, lung cancer, chondrosarcoma, and germ cell tumor cell lines, without any side effects on normal human cells [80,82,83,84,85].

Many compounds from natural products are now considered to be potential chemotherapeutic agents for the treatment of cancer. Coincidentally, a large number of studies have indicated that various natural compounds can reduce telomerase activity and hTERT mRNA/protein levels, which is accompanied by DNA damage and apoptosis in various cancer cell lines [86,87]. Here, two such compounds are briefly described. Curcumin, which is a compound derived from turmeric, has been shown to inhibit hTERT expression and decrease the expression of both mRNA and protein, resulting in telomerase dysfunction in various human cancer cell lines [88,89]. Another study reported that the anticancer capability of curcumin can treat cancer by altering telomerase activity, inducing senescence-like cell cycle arrest, triggering apoptosis, and decreasing cell proliferation [90]. Moreover, curcumin has also been shown to block the association of p23 and the hTERT complex then disrupted the assemble of telomerase, resulting in telomerase inactivation and cell death in cancer cells [88]. The other example is the bioactive constituent in green tea, (-)-epigallocatechin-3-gallate (EGCG), which was shown to downregulate the expression of both hTERT mRNA and protein, leading to cell cycle arrest in both G2/M and S phases, and promoting DNA damage responses and apoptosis in cancer cell lines [91,92,93,94,95].

In addition to these small molecular inhibitors, there are other therapeutic approaches that inhibit telomerase functioning, such as immunotherapy. GRNVAC1/2, GV1001, GX301, and VX-001 are typical anticancer vaccines that are currently in clinical trials. These anticancer vaccines are designed to stimulate the CD4+ and CD8+ immune response to hTERT antigens and lead to tumor cell lysis by harnessing the immune systems of patients. Some previous preclinical and clinical studies demonstrated that hTERT-based peptide vaccines can inhibit cancer by inducing immunological activation in several types of tumor cells, including breast, colorectal, ovarian, head and neck, pancreatic, hepatocellular, renal, melanomas, osteosarcomas, and prostate cancer cells [96,97,98,99].

### 4.2. Inhibition of Telomere Extension by Anticancer Drugs

Another strategy used to inhibit telomerase involves targeting telomeres. The 3′ overhang in the telomere is a guanine-rich region and adopts a 4-strand DNA structure known as a G-quadruplex, which is controlled precisely to allow DNA replication and cell division [100]. Previous studies have indicated that G-quadruplexes obstruct telomerase activity and telomere extension [101,102,103]. G-quadruplexes can be resolved only by DNA helicases and not by telomerase [104,105]. Thus, the binding and stabilizing of telomeric G-quadruplexes by G-quadruplex-specific small molecules is a potential anticancer therapeutic strategy that could be used to alter telomere function and inhibit cancer cell proliferation.

In the past decade, many G-quadruplex ligands have been synthesized. BRACO-19, RHPS4, and porphyrin (TMPyP4) are some of the most attractive that have been developed to increase G-quadruplex stability [106,107,108,109]. Experimental data have proven that these ligands binds with high affinity to 3′ end of the telomere prompt the formation of G-quadruplexes in some malignant cancers. Additional studies have demonstrated that treatment with these compounds can shorten the 3′ overhang and result in a DNA damage response in tumor cells [110]. In various experimental models, those ligands also resulted in significant tumor reduction. Research has also shown that BRACO-19 not only blocks telomere capping activity but also may cause partial telomerase inhibition. The data suggested that hTERT is colocalized with ubiquitin and undergoes proteolysis due to BRACO-19 treatment [111,112]. Similarly, CX-5461 and CX-4945 have been shown to be effective G-quadruplex-interacting agents, and both are in phase I/II clinical trials and are highly regarded by cancer experts [113]. 

As expected, some anticancer natural products are also G-quadruplex stabilizers. For example, the macrocyclic peptide isolated from *Streptomyces anulatus*, telomestatin, is one of the most potent G-quadruplex binding small molecules, which is promising for cancer chemotherapy [114]. Because of its specific selective binding to G-quadruplexes, telomestatin can lead to telomere degradation and induce delayed senescence and apoptosis in various types of cancer cells, such as leukemia, neuroblastoma, and cervical cancer cells, while having minimum effects on normal cells.

### 4.3. Inhibition of Telomerase Complex-Related Proteins Could Be a Promising Anticancer Strategy

Apart from the two major components, the telomerase complex contains several other proteins, such as heat shock protein 90 (HSP90), p23, tankyrase (TNKS), and dyskerin [115,116]. Here, we describe some telomerase-related protein-based therapeutic strategies. First, heat shock protein 90 (HSP90), which is a molecular chaperone that interacts with its cochaperone p23, is responsible for protein folding and the preventing of misfolding [117]. Since HSP90 regulates tumorigenic proteins, the inhibition of HSP90 signaling pathways in tumors becomes a promising target for anti-angiogenesis and anti-tumorigenesis. In fact, the HSP90-p23 chaperone complex is necessary for the maturation and activation of telomerase [118,119,120]. These two chaperone proteins are involved in the assembly of the catalytic domains of telomerase on their RNA templates. One study found that hTERT gene promoter activity requires hTERT-HSP90 interaction in cancer cells [121]. Other studies demonstrated that the inhibition of HSP90 could lead hTERT to be ubiquitinated and degraded by the proteasomal pathway, and reduce DNA extension by telomerase [122,123]. Geldanamycin (GA), which is benzoquinone ansamycin (BA) antibiotic, is a classical HSP90 inhibitor that acts by binding to the unusual ADP/ATP-binding pocket of HSP90 [124]. 

Another important component of the telomerase complex is TNKS, which comprises the poly (ADP-ribose) polymerase (PARP) family members tankyrase 1 and 2 (TNKS1 and TNKS2) [125]. Tankyrases are telomerase-specific PARPs that can interact with target proteins and regulate a variety of cellular processes, including telomere maintenance. The telomeric DNA-binding proteins include telomeric repeat binding factor 1 (TRF1) and telomeric repeat binding factor 2 (TRF2), which are in the sheltering complex and are required for formation of the t-loop and the telomerase complex. They form a cap on telomere ends and regulate telomere length by inhibiting telomerase [126]. However, TNKS1 can induce poly ADP-ribosylation of TRF1, which dissociates TRF1 from telomeric DNA, and results in the exposure of the telomere to telomerase and the promotion of telomere elongation [127,128,129]. Thus, the targeting tankyrases by PARP inhibitors may be a strong anticancer strategy [130]. 

Dyskerin is another important protein involved in the telomerase complex that is required for both ribosome biogenesis and telomerase complex stabilization. Dyskerin is overexpressed in many human cancer cell types, so it is suitable as a novel treatment for cancer [131,132]. Recently, the Wang’ group tested a series of new trimethoxyphenyl-4H-chromen derivatives as telomerase inhibitors. These compounds showed great anticancer effects and inhibited cancer cell growth via the induction of apoptosis. The data indicated that these derivatives could decrease the expression of dyskerin, which was followed by the induction of apoptosis, the inhibition of telomerase activity, and the arrest of the cell cycle due to dyskerin inhibition [133].

## 5. Anticancer Effects of Stilbene Compounds Are Mediated by the Targeting of Telomerase and Cellular Senescence

### 5.1. Classification of Stilbene Compounds and Their Anticancer Mechanisms

Stilbene compounds are part of a group of polyphenols that occur in many plant species, such as grape, blueberry, peanut, almond, and many tree species (*Pinus* and *Picea*) [134,135]. Stilbene can also be found in folk medicines used in Asia, such as those derived from *Pterocarpus indicus*, *Polygonum cuspidatum*, *Rhodomyrtus tomentosa*, *Rheum undulatum*, *Melaleuca leucadendron*, and *Euphorbia lagascae* [135]. This phenomenon suggests that important sources of stilbenes include food and folk medicines and have nutraceutical applications [135]. Stilbenes are phytochemicals with 200–300 g/mol molecular weight, a subclass of polyphenolic compounds. The stilbene family includes resveratrol (3,5,4′-trihydroxy-trans-stilbene), pterostilbene (trans-3,5-dimethoxy-4-hydroxystilbene), piceatannol (3,4′,3′,5-trans-trihydroxystilbene), hydroxystilbene (2-(2-phenylethenyl)phenol), amorphastilbol (3,5-dihydroxy-4-geranylstilbene), pinostilbene (3,4’-dihydroxy-5-methoxystilbene), rhapontigenin (3,3’,5-trihydroxy-4’-methoxystilbene), and isorhapontigenin (trans-3,5,4′-trihydroxy-3′-methoxystilbene) [136]. Most of these compounds have beneficial health effects, including antioxidative, anti-inflammatory, neuroprotective, cardiovascular protective, antiaging, and chemopreventive effects [135]. In terms of molecular mechanisms, evidence has indicated that the anticancer effects of stilbene compounds include the enhancement of apoptosis, senescence, autophagy, and cell cycle arrest, and the inhibition of the invasion and metastasis of cancer cells [137,138,139]. As mentioned above, cancer senescence can be triggered by various stimuli such as ROS production, DNA damage, telomerase inactivation, and permanent cell cycle arrest. This suggest that stilbene compounds have the potential to induce senescence through one or many of these mechanisms. However, the studies of anti-cancer mechanisms of stilbenes targeting to senescence or telomerase remain limited so far. In this section, we describe the most well-studied stilbene compounds that have been shown to induce senescence and inhibit telomerase activity in cancer cells. The mechanisms and the outcomes of the studies are provided in Table 1 and are schematically represented in Figure 1.

### 5.2. Anticancer Mechanisms of RSV, RSV Derivatibes, or Combined Therapy by Targeting to Telomerase and Cellular Senescence

Among the stilbene compounds, trans-resveratrol, also known as 3,4,5,’-tri-hydroxystilbene, and pterostilbene (trans-3,5-dimethoxy-4-hydroxystilbene, which is a dimethylether analog of resveratrol), have been well-studied as promising dietary polyphenolic compounds with various chemopreventive and chemotherapeutic effects [134,140]. Previous studies indicated that the regulation of telomerase activity, telomere structure, and telomerase- and telomere-associated binding proteins have been suggested as new avenues for therapeutic intervention for some tumors in which it is difficult to induce apoptosis, such as glioblastoma (GBM) [140]. Mirzazadeh et al. indicated that resveratrol (RSV) inhibited hTERT mRNA expression and cell proliferation in U-87 MG cells [140]. In a study by Kim et al., the stress hormone norepinephrine (NE) augmented cancer cell proliferation via the β2-adrenergic receptor and induced hTERT activity via Slug, which subsequently increased ovarian cancer cell invasion [143]. Kim et al. further indicated that RSV downregulates NE-induced hTERT expression and subsequent Slug expression by inhibiting Src phosphorylation and HIF-1α expression in ovarian cancer cells [143]. Similar results were reported that showed that RSV reduced hTERT expression and c-Myc expression in colon cancer cells [141]. 

RSV is one of the most active natural products in inducing senescence in cancer cells, particularly at concentration equal to or lower than 50 μM [13]. The mechanism used by RSV to induce senescence in cancer cells was reported to involve the p53, p21^CIP1/WAF1^, p16^INK4a^, and SIRT1 pathways [157]. Previous studies have suggested that RSV-induced senescence requires p53 activation. Upregulation of the p53-CXCR2 axis is largely responsible for RSV-induced senescence. Importantly, Li et al. indicated that CXCR2 drives senescence but acts as a barrier to apoptosis [158]. In addition, S phase arrest is commonly observed in cells treated with RSV or PT to induce senescence, reflecting that replication stress was induced when replication forks were stalled [134,158]. In MCF-7 breast cancer cells, Lanzilli et al. indicated that RSV induced S phase arrest that is associated with the inhibition of hTERT activity and protein expression [142]. The induction of oxidative stress was also reported to mediate RSV-induced senescence in HCT116 cells [159]. Interestingly, in U2OS cells, Li et al. indicated that oxidative stress acted as a mediator of senescence after cells had already experienced replication stress [158]. The study indicated that both oxidative stress and replication stress induced senescence in response to RSV [159]. 

However, due to the low bioavailability and fast metabolization of RSV, the clinical value of RSV is limited [145]. Therefore, numerous modified derivatives of RSV have been synthesized, and their activity has been evaluated. For instance, in a study by Martí-Centelles, several stilbene derivatives were prepared, and their cytotoxic activity was evaluated [144]. (E)-4-(4-methoxystyryl) aniline showed the highest cytotoxicity and greatest hTERT inhibition activity when compared with other stilbenes [144]. Another RSV derivative, 3,3’,4,4’,5,5’-hexahydroxy-trans-stilbene (M8), which was prepared by Mikuła-Pietrasik, has been found to inhibit growth or induce apoptosis in various cancers, including breast cancer, colon cancer, leukemia, melanoma, and glioma [145]. M8 was reported to be even stronger than RSV due to the ortho-hydroxyl groups in its structure, which induced the production of mitochondrial reactive oxygen species (ROS) to exert antiproliferative and proapoptotic activities against cancers [160]. Moreover, M8 could accelerate the induction of senescence via an oxidative stress-dependent mechanism in human peritoneal mesothelial cells (HPMCs) [145]. Recent studies have indicated that RSV prodrugs could serve as an alternatives to increase RSV bioavailability [146]. It has been reported that RSV prodrugs modified with glucosyl or acyl groups inhibit the transcription of hTERT and cMyc in HT29 colon cancer cells [161]. These studies indicated that RSV prodrugs such as resveratrol-3,5-diglucoside or RSV sulfate metabolites showed equal or higher cytotoxicity than RSV in HT-29 and MCF-7 cells [146]. The exact mechanisms of hTERT inhibition remained unclear, but the author suggested the underlying mechanisms were different from those of RSV [146].

Other studies indicated that RSV synergistically induced cytotoxicity in cancer cells in combination therapy. Chung et al. indicated that RSV enhanced the antitelomeric activities of 5-FU by inhibiting Stat3 binding to the hTERT promoter region [151]. Fang et al. established a combination model using RSV and radiation therapy (XRT) to treat prostate cancer [147]. The results showed that the inhibition of PC-3 PCA cell by XRT/RSV growth was mediated by the increased expression of p-H2AX and p21^CIP1/WAF1^, p27^Kip1^, and that p53 is also involved and might contribute to senescence in prostate cancer cells [147]. Similarly, RSV enhanced ionizing radiation (IR)-induced senescence in lung cancer cells through ROS-mediated pathways [148]. In the treatment of GBM, temozolomide (TMZ) is the most widely used drug that induces G2 cell cycle arrest and extensive DNA damage responses to cause apoptosis, mitotic catastrophe, autophagy, and senescence-like events in cells [152]. In a study by Filippi-Chiela, RSV potentiated the cytotoxic effect of TMZ on GBM by increasing DNA damage responses (DDR), which led to metabolic catastrophe and, in the long term, to senescence and a reduction in clonogenic survival [152]. Interestingly, the effect of RSV in increasing TMZ-induced toxicity and autophagy in glioma cells is p53-independent. In rat glioblastoma C6 cells, low doses of RSV combined with another flavonoid known as quercetin, which is also a potential anticancer agent, synergistically induced long-term senescence-like reduction in cell growth [149]. Overall, previous studies have shown that treatment with RSV alone or in combination triggered multiple cell death mechanisms, including senescence and hTERT inhibition, and thus reduced the possibility of the resistance and recurrence of cancer cells.

### 5.3. Anticancer Mechanisms of PT Are Mediated by the Targeting of Telomerase and Cellular Senescence

Many studies have shown that PT exhibits high bioavailability and a high potential for cellular uptake. Moreover, PT has a longer half-life and better biological effects than RSV [149,162]. Therefore, PT has attracted more attention than RSV due to its better pharmaceutical characteristics. Studies regarding hTERT inhibition and the senescence-inducing effects of PT have been limited. Tippani et al. first performed molecular docking studies to evaluate the interaction between PT and the active site of telomerase [153]. The study indicated that PT has a strong interaction with the telomerase’ active site and has a docking energy of −7.10 kcal/mol [153]. The study further showed that PT significantly inhibited telomerase activity in MCF-7 and NCI H-460 lung cancer cells at a concentration of 80 μM [153]. Lee et al. reported that, compared with other stilbene compounds, such as rhapontigenin, piceatannol, and RSV, PT showed the greatest cytotoxicity in NSCLC lung cancer cells and exerted its effects in a p53-dependent manner [154]. Low doses of PT induced replicative stress, resulting in cell cycle arrest at the S phase and the activation of the ATM-CHK-p53 axis, leading to senescence [154]. Similarly, Daniel et al. indicated that PT arrested MCF-7 cells in the G1 phase and MDA-MB-231 cells in the G2/M phase, which is associated with p53 status, and showed that hTERT expression was decreased in both types of cancer cells after PT treatment [155]. In addition, cMyc, which is a proto-oncogene and an activator of hTERT, was downregulated by PT [155]. However, silenced cMyc could not completely inhibit the transcriptional activity of hTERT, suggesting that other forms of transcriptional regulation existed [155]. Consistently, our previous study also showed that PT effectively induced senescence in lung cancer cells at a concentration of 50 μM [134]. We further investigated how PT induced replicative stress, and the results showed for the first time that PT inhibits hTERT enzymatic activity and protein expression, resulting in the subsequent induction of DNA damage, the activation of ATM/Chk2 and p53, and S phase arrest. The activation of p53-related positive feedback provokes hTERT downregulation, resulting in senescence in H460 cells. Interestingly, PT only slightly inhibited hTERT activity in H1299 cells, which reduced cell senescence, suggesting that PT-induced senescence in lung cancer cells partially occurs via p53-mediated hTERT inhibition [134]. Therefore, previous studies, including ours, have suggested that PT inhibits hTERT expression and activity, leading to replication stress, S phase arrest, and senescence that depends on the status of p53 in lung cancer cells [134].

It is now known that cancer is not a single disease, and the idea of a “monoshot” therapeutic strategy is misleading and outdated [140]. Therefore, we propose that the use of stilbene compounds, especially PT, for cancer therapy may be beneficial because PT activity is not limited to the regulation of hTERT activity and expression but is also involved in the downregulation of complex survival pathways in cancer cells. In accordance with other results, we have pointed out the key role of telomerase in carcinogenesis and encourage scientists to consider that telomerase could be a fundamental marker for cancer therapy. Telomeres and telomerase are attractive targets for anticancer therapy. The regulation of telomerase activity can be achieved at either the transcriptional level or via posttranslational alternative splicing of hTERT [134,142]. In this regard, RSV and PT, two of the most effective stilbenes, may potentially be used as therapeutic agents for cancers with high telomerase activity and increased hTERT expression. In addition, hTERT inhibition could induce senescence in cancer cells, and the clearance of senescent cells by immune cells [13]. Considering the safety profiles of stilbene compounds, the majority of studies have indicated that RSV and PT are well tolerated and do not produce marked toxicity [151]. For instance, the results of clinical trials showed that PT is generally safe for use in humans at amounts up to 250 mg/day and is also safe for treating or preventing cardiovascular disease [161]. In light of the safety profiles in humans and the beneficial chemopreventive effects of stilbenes, it is suggested that stilbene compounds can potentially be used for chemotherapy.

## 6. Conclusion and Perspectives

Robust scientific evidence demonstrates that stilbene compounds are capable of suppressing oxidative stress and inflammation as well as modulating senescence, apoptosis, autophagy, and cell cycle arrest. Cancer senescence can be triggered by various stimuli such as DNA damage, telomerase inactivation, and permanent cell cycle arrest. This suggest that stilbene compounds have the potential to induce senescence through the inhibition of telomerase activity in cancer cells. The complex relationship between telomeres/telomerase at the different levels of the regulation of telomerase activity and their relationship with associated proteins suggests that these associated proteins could be excellent targets in the fight against cancer [163]. However, most studies have focused on RSV, and only limited research has been carried out with other stilbene compounds, such as PIC and PT, which may have higher biological activity and deserve more scientific attention [135]. 

The tumor-suppressive function of senescence provides an opportunity for the development of treatments that enhance senescence in clinical cancer therapy. Many of the currently-used chemotherapeutic drugs act by inducing severe DNA damage and triggering cellular senescence in tumor cells, which could contribute to the success of chemotherapy [15]. Therapies that enhance senescence not only promote stable cell growth arrest but also function as a potent stimulus for the activation of antitumor immune responses. Although the main role of senescence is thought to be related to tumor suppression, detailed studies are needed to characterize the exact role of senescence in cancer. Moreover, a deeper understanding of the molecular mechanisms that determine tumor-suppressive effects will help in the development of senescence-targeting therapies. In conclusion, the use of natural compounds and chemotherapeutic drugs, either alone or as part of combined therapies, with the aim of inducing senescence in cancer cells appears to be a promising therapeutic strategy to treat cancer.

## Figures and Tables

**Figure 1 ijms-20-02716-f001:**
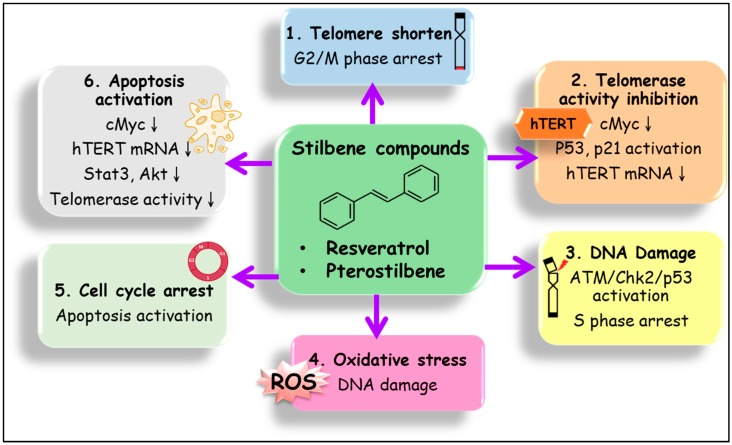
Anticancer mechanisms of stilbene compounds by targeting of the telomerase reverse transcriptase (hTERT) and senescence related pathways. The core structure of stilbene family is 1,2-diphenylethylene based skeleton. The derivative of stilbenes compounds such as resveratrol (RSV, 3,4′,5-trihydroxy-trans-stilbene), pterostilbene (PT, trans-3,5-dimethoxy-4-hydroxystilbene), and piceatannol (PIC, 3,4′,3′,5-trans-trihydroxystilbene), can inhibit cancer cell proliferation by inhibiting hTERT expression/activity and inducing cellular via various pathways, including (**1**) telomere shortening leading to G2/M arrest, (**2**) telomerase activity inhibition by hTERT mRNA express inhibition, p53/p21 activation and cMyc inhibition, (**3**) DNA damage and subsequently ataxia-telangiectasia mutated (ATM)/Chk2/p53 activation and S phase arrest, (**4**) cellular oxidative stress induced DNA damage, (**5**) cell cycle arrest and apoptosis activation, and (**6**) apoptosis activation by inhibition of cMyc, hTERT mRNA express, Stat3/Akt, and telomerase activity.

**Table 1 ijms-20-02716-t001:** Stilbene compounds and their anticancer mechanisms by inhibition of telomerase and induction of senescence.

Stilbene Compounds	Cell Models	Target/Mechanism	Outcome	Reference
**Resveratrol**				
RSV	U-87MG	hTERT mRNA↓	cell growth↓, cell death↑	[140]
RSV	HT-29, WiDr	Telomerase activity↓	cell proliferation↓	[141]
RSV	MCF-7	Telomerase activity↓ hTERT expression↓	S phase arrest↑ apoptosis↑	[142]
RSV	Ovaria cancer cells	hTERT expression↓ Slug↓, pScr and HIF-1α↓	EMT↓ invasion↓	[143]
**Resveratrol Derivatives**				
RSV derivative (E)-4-(4-methoxystyryl) aniline	Colon cancer cells	hTERT expression↓ cMyc expression↓	cytotoxicity↑	[144]
RSV derivative 3,3’,4,4’,5,5’-Hexahydroxy-trans-Stilbene (M8)	Human peritoneal mesothelial cells	mitochondrial reactive oxygen species↑	senescence↑, cell cycle arrest↑	[145]
RSV sulfate metabolites	HT29, MCF-7	hTERT mRNA↓, cMyc ↓	cell death↑	[146]
**Resveratrol Combined Therapy**				
RSV + XRT	Prostate cancer cells	p21^CIP1/WAF1^, p27^Kip1^, p53↑ Fas, TRAIL1↑, p-H2AX↑	senescence↑ apoptosis↑	[147]
RSV + IR	Lung cancer cells	ROS↑, DDR↑	senescence	[148]
RSV + quercetin	Rat glimoma cells	caspase 3/7 activation↑	senescence	[149]
RSV + 5-FU	Colon cancer cells	Telomerase activity↓ Stat3 and Akt↓	apoptosis	[150,151]
RSV + TMZ	Glioblastoma cells	Mitotic catastrophe p-ATM, p-Chk2↑	senescence↑ autophagy↑	[152]
**Pterostilbene**				
PT	Lung cancer cells	Molecular docking to hTERT	cell death	[153]
PT	Lung cancer cells	ATM-CHK-p53	senescence↑	[154]
PT	Breast cancer cells	cMyc expression↓ hTERT expression↓	apoptosis cell cycle arrest	[155]
PT	Lung cancer cells	hTERT activity↓, hTERT expression↓, DDR↑, p53-dependent	S phase arrest↑ senescence↑	[134]
PIC	Hepatic stellate cell	p16^INK4a^↑, p53↑ Bcl-xl↓, SMAD↓	Inflammation↓ hepatocarcinogenesis↓	[156]

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
