# Peer review of "Stilbene Compounds Inhibit Tumor Growth by the Induction of Cellular Senescence and the Inhibition of Telomerase Activity"

_ijms, 2019, doi:10.3390/ijms20112716_

Reviewer 1 Report

This is a nicely written review by Lee et al. summarizing the interplay between stilbene activity, antitumor effects and cellular sentence and the role of telomerase.  Generally the review is well structured and logically to follow. It gives an overview of theses family of natural compounds in the light of senescence and telomere stability.

Minor points:

The first part is sometimes a bit repetitive and lengthy. For example the inducers of senescence are mentioned again and again in several subchapters. The authors should try to streamline the text here.

Line 43-44: Telomerase is a prominent target but it should be mentioned that no directs telomerase inhibitor has been approved for clinical use so far.

Line 108-109: The sentence does not make any sense in the present form: “In the absence RAS overexpression, stressors drive cells into senescence, and this mechanism works as a barrier to block tumor growth in vivo [21].”

Line 131-133: It needs to be mentioned that taxol is primarily targeting microtubules and the mitotic spindle.

Line 330-332: Rewrite the sentence to “BRACO-19, RHPS4 and porphyrin (TMPyP4) are some of the most attractive ligands that have been developed to increase G-quadruplex stability [125-128].”

Please explain that in figure 1 the chemical formula of stilbene is shown and not of resveratrol or pterostilbene. As the names are directly below the formula this might be misleading.

Author Response

Comments and Suggestions for Authors

This is a nicely written review by Lee et al. summarizing the interplay between stilbene activity, antitumor effects and cellular sentence and the role of telomerase. Generally the review is well structured and logically to follow. It gives an overview of theses family of natural compounds in the light of senescence and telomere stability.

Comment:

Minor points:

The first part is sometimes a bit repetitive and lengthy. For example, the inducers of senescence are mentioned again and again in several subchapters. The authors should try to streamline the text here.

Response:

Thank you for the suggestion. We have checked the text and streamlined all subchapters about senescence. Please refer to the revised manuscript.

Comment:

Line 43-44: Telomerase is a prominent target but it should be mentioned that no directs telomerase inhibitor has been approved for clinical use so far.

Response:

Thank you for the comments. We have re-written these sentences shown as following: Since the discovery of telomerase, which is an enzyme that is able to elongate telomeres and is mostly active in tumors, it has become a prominent target of therapy. Although different methods of inhibiting telomerase have been attempted in the preclinical studies, however, no directs telomerase inhibitor has been approved for clinical use so far.

Comment:

Line 108-109: The sentence does not make any sense in the present form: “In the absence RAS overexpression, stressors drive cells into senescence, and this mechanism works as a barrier to block tumor growth in vivo [21].”

Response:

Thank you for the comments. We have corrected the sentence: In the absence apoptosis, stressors could also drive cells into senescence through RAS, and this mechanism works as a barrier to block tumor growth in vivo.

Comment:

Line 131-133: It needs to be mentioned that taxol is primarily targeting microtubules and the mitotic spindle.

Response:

Thank you for the suggestion, and we have re-written these sentences shown as following: Paclitaxel is the first microtubule stabilizing agent that could suppresses spindle microtubule dynamics resulted in the inhibition of mitosis and induction of apoptosis in cancer cells. In addition, paclitaxel and its water-soluble conjugates can inhibit tumorigenesis by inducing extensive telomere erosion [29].

Comment:

Line 330-332: Rewrite the sentence to “BRACO-19, RHPS4 and porphyrin (TMPyP4) are some of the most attractive ligands that have been developed to increase G-quadruplex stability [125-128].”

Response:

Thank you for the suggestion and the sentence has been corrected.

Comment:

Please explain that in figure 1 the chemical formula of stilbene is shown and not of resveratrol or pterostilbene. As the names are directly below the formula this might be misleading.

Response:

Thank you for the comments. We have added the explanation of the formula of stilbene compounds in Figure 1 legends shown as following: “The core structure of stilbene family is 1,2-diphenylethylene based skeleton. The derivative of stilbenes compounds such as resveratrol (RSV, 3,4’,5-trihydroxy-trans-stilbene), pterostilbene (PT, trans-3,5-dimethoxy-4-hydroxystilbene) and piceatannol, (PIC, 3,4’,3’,5-trans-trihydroxystilbene), ……”.

Reviewer 2 Report

In the review " Stilbene Compounds Inhibit Tumor Growth by the Induction of Cellular Senescence and the Inhibition of Telomerase Activity" the authors present the role, signaling pathways, and the targeting of senescence and telomerase in anticancer therapy, as well as the implication of stilbenes in these mechanisms. In the present form, the manuscript structure it's not in accordance with the title, the stilbene compounds section (Section 4) being poorly presented. I suggest to the authors a different approach of the theme, such that the content of the review to be in agreement with the title and keywords. Thus, I consider that section 4 should be significantly changed and re-organized in two or three sections. These sections should present clearly the classification of these phytochemicals (1), their implication in the modulation of other pathways which are involved in tumor cells and the possible interaction with the senescence and telomerase mechanisms (2) and the state of the art regarding their implications in senescence and telomerase (3). The conclusion should be re-organized, especially the second paragraph (rows 142 - 152/ page 15).

Minor

rows 142-143, 248, 290, 301 - 305: Please correct the phrases.

rows 180 - 185 Please rephrase.

Author Response

In the review " Stilbene Compounds Inhibit Tumor Growth by the Induction of Cellular Senescence and the Inhibition of Telomerase Activity" the authors present the role, signaling pathways, and the targeting of senescence and telomerase in anticancer therapy, as well as the implication of stilbenes in these mechanisms. In the present form, the manuscript structure it's not in accordance with the title, the stilbene compounds section (Section 4) being poorly presented. I suggest to the authors a different approach of the theme, such that the content of the review to be in agreement with the title and keywords. Thus, I consider that section 4 should be significantly changed and re-organized in two or three sections. These sections should present clearly the classification of these phytochemicals (1), their implication in the modulation of other pathways which are involved in tumor cells and the possible interaction with the senescence and telomerase mechanisms (2) and the state of the art regarding their implications in senescence and telomerase (3). The conclusion should be re-organized, especially the second paragraph (rows 142 - 152/ page 15).

Response:

Thank you for the suggestion. We have re-organized and rephrased section 4 and conclusion. The brief description of classification of stilbene compounds and their anticancer mechanisms that are possible interaction with the senescence or telomerase have been added in the section: 4.1 Classification of stilbene compounds and their anticancer mechanisms. The anticancer mechanisms of RSV and PT by targeting to senescence and telomerase have been discussed separately because they have different anticancer mechanisms and different therapeutic strategies regarding to senescence or telomerase. Please refer to all the reorganized sections in the revised version.

Minor comments:

rows 142-143, 248, 290, 301 - 305: Please correct the phrases.

rows 180 - 185 Please rephrase

Response:

Thank you for the comment. These sentences are corrected and rephrased accordingly.

Reviewer 3 Report

The manuscript by Lee et al. claims to review the anticancer-related activity of stilbene compounds with the emphasis on its ability to induce the cellular senescence and to inhibit the telomerase activity. At the same time more then half of the current review article is devoted to the senescence itself as well as it's mechanisms. Thus, in my opinion the manuscript either has to be rearranged or the it's title (chapter titles) has to be revised accordingly. Additionally the abbreviation section should be introduced to make the manuscript more readable. Apart from it the manuscript is of good quality and well written, and therefore it could be accepted in the jounal after some minor corrections.

Author Response

Comments and Suggestions for Authors

The manuscript by Lee et al. claims to review the anticancer-related activity of stilbene compounds with the emphasis on its ability to induce the cellular senescence and to inhibit the telomerase activity. At the same time more then half of the current review article is devoted to the senescence itself as well as it's mechanisms. Thus, in my opinion the manuscript either has to be rearranged or the it's title (chapter titles) has to be revised accordingly. Additionally the abbreviation section should be introduced to make the manuscript more readable. Apart from it the manuscript is of good quality and well written, and therefore it could be accepted in the jounal after some minor corrections.

 Response:

Thank you for the suggestion. We have revised the chapter titles and added the abbreviation section as an independent paragraph after author contributions. Please refer to the revised version.

Round  2

Reviewer 2 Report

The manuscript has been significantly improved and it's suitable for publication in IJMS.